# Comparison of lncRNA Expression in the Uterus between Periods of Embryo Implantation and Labor in Mice

**DOI:** 10.3390/ani12030399

**Published:** 2022-02-08

**Authors:** Zijiao Zhao, Lu Chen, Maosheng Cao, Tong Chen, Yiqiu Huang, Nan Wang, Boqi Zhang, Fangxia Li, Kaimin Chen, Chenfeng Yuan, Chunjin Li, Xu Zhou

**Affiliations:** College of Animal Sciences, Jilin University, Changchun 130062, China; zhaozj18@mails.jlu.edu.cn (Z.Z.); luchen@jlu.edu.cn (L.C.); Caoms18@mails.jlu.edu.cn (M.C.); chentong19@mails.jlu.edu.cn (T.C.); 17843103601@163.com (Y.H.); nanwang19@mails.jlu.edu.cn (N.W.); zhangbq19@mails.jlu.edu.cn (B.Z.); lifx9917@163.com (F.L.); ckmjy134@163.com (K.C.); yuancf20@mails.jlu.edu.cn (C.Y.)

**Keywords:** P4, embryo implantation, labor, mRNAs, lncRNAs

## Abstract

**Simple Summary:**

Progesterone has been proven to play an important role in female mammals during pregnancy. In this study, the uteri of pregnant mice were collected to compare mRNA and lncRNA expression between periods of embryo implantation and labor. The results show that 19 known differentially expressed lncRNAs and 31 novel differentially lncRNAs were commonly expressed between the two stages, indicating that these lncRNAs’ function is related to progesterone.

**Abstract:**

Uterine function during pregnancy is regulated mainly by progesterone (P4) and estrogen (E2). Serum P4 levels are known to fluctuate significantly over the course of pregnancy, especially during embryo implantation and labor. In this study, pregnant mice at E0.5, E4.5, E15.5, and E18.5 (*n* = 3/E) were used for an RNA-Seq-based analysis of mRNA and lncRNA expression. In this analysis, 1971 differentially expressed (DE) mRNAs, 493 known DE lncRNAs, and 1041 novel DE lncRNAs were found between E0.5 and E4.5 at the embryo implantation stage, while 1149 DE mRNAs, 192 known DE lncRNAs, and 218 novel DE lncRNAs were found between E15.5 and E18.5 at the labor stage. The expression level of lncRNA-MMP11 was significantly downregulated by P4 treatment on MSM cells, while lncRNA-ANKRD37 was significantly upregulated. Notably, 117 DE mRNAs, 19 known DE lncRNAs, and 31 novel DE lncRNAs were commonly expressed between the two stages, indicating that these mRNAs and lncRNAs may be directly or indirectly regulated by P4.

## 1. Introduction

In female mammalian reproduction, crosstalk between the embryo and the uterus is present throughout pregnancy. Crosstalk becomes particularly frequent during the embryo implantation stage and labor stage, and is regulated mainly by progesterone (P4) and estrogen (E2), as well as other hormones, such as oxytocin and human chorionic gonadotropin [1,2,3,4,5,6,7,8]. P4 has been proven to play an important role in embryo implantation and labor [9,10,11,12,13]. Implantation and labor are complex processes involving paracrine, autocrine, cell–cell and cell–matrix interactions [4,14,15,16,17]. The mouse is the preferred model animal for implantation and labor research that focuses on changes in uterine morphology and physiology [14,18]. The levels of P4 remain low until the end of implantation at embryonic day (E) 4.5 [19]. At this stage, the uterine environment is beneficial to support blastocyst growth and implantation [20,21,22]. The high levels of P4 were treated as the main hormone to regulate myometrial quiescence throughout the majority of pregnancy [11,23,24]. The P4 level remains high to maintain uterine quiescence until E15.5, with progesterone receptor function decreasing (Figure 1A) [25,26,27]. Both stages are connected with an increased level of inflammatory response; this is characterized by increased production of proinflammatory cytokines such as IL-1β and IL6 (Figure 1B) [23]. Although embryo implantation and labor are different physiological stages, the stages may be regulated by changes in P4 via a similar molecular mechanism.

Long noncoding RNAs (lncRNAs) are important components of noncoding RNAs [28]; lncRNAs can regulate target genes by cis or trans, or in combination with miRNAs, so it is difficult to determine their target genes [29,30]. The roles of lncRNAs have been widely reported in organ development, cancer, and aging over recent decades [31,32,33]. LncRNAs can also be biomarkers of endometrial receptivity in humans [34]. Therefore, this study was conducted in order to further explore the expression of lncRNAs that might be regulated by P4. The uteri of pregnant mice at embryonic day 0.5 (E0.5), embryonic day 4.5 (E4.5), embryonic day 15.5 (E15.5), and embryonic day 18.5 (E18.5) were collected in order to analyze their lncRNA and mRNA expression profiles (Figure 1C). In mice, fertilization is completed in the fallopian tube on E0.5, and endometrial receptivity lasts for 18–24 h, or until the end of 5 days post-coitum [35,36]. Thus, in this study, E0.5 and E4.5 were considered to represent the embryo implantation stage according to P4 levels, and E15.5 and E18.5 represented the labor stage according to P4 levels and other parturition studies [14,19,37,38,39,40]. This is the first study to show that differentially expressed (DE) lncRNAs and mRNAs in the murine uterus might be regulated by P4, and we suggest that this will provide basic data for exploiting the function of lncRNAs in murine embryo implantation and labor.

## 2. Methods

### 2.1. Animals and Sample Collection

Twelve female ICR mice were purchased from Liaoning Changsheng Company (Shenyang, China). At the age of 60 days, they were mated with male mice to become pregnant. The 12 mice were randomly divided into 4 groups, with 3 mice in each group. On days 0.5 (E0.5; m0051, m0052, m0053), 4.5 (E4.5; m0451, m0452, m0453), 15.5 (E15.5; m151, m152, m153), and 18.5 (E18.5; m181, m182, m183) after mating, the whole uterus of each mouse was collected. All uteri were immediately snap-frozen in liquid nitrogen and stored at −80 °C.

### 2.2. Total RNA Isolation, Library Preparation, and Sequencing

The tissues of the four groups were sequenced by a service provider (LC-BIO Biotech Ltd., Hangzhou, China). Total RNA was isolated and purified from each uterine sample using TRIzol reagent (Invitrogen, Carlsbad, CA, USA). DNase I (Takara, Japan) was used to eliminate DNA contamination. The RNA concentration and integrity of individual samples were estimated using a NanoDrop ND-1000 (NanoDrop, Wilmington, DE, USA). Total RNA was depleted of ribosomal (rRNA) molecules using the Ribo-Zero rRNA Removal Kit (Illumina, San Diego, CA, USA). Divalent cations were used to fragment the remaining RNA into small fragments. Then, the cleaved RNA fragments were reverse-transcribed to produce cDNA. The size of cDNA fragments in the library was ~300 bp (±50 bp). Finally, paired-end sequencing was performed on an Illumina HiSeq 4000 (LC-BIO Biotech Ltd., Hangzhou, China).

### 2.3. Quality Control and Mapping

First, Cutadapt (1.9) (https://cutadapt.readthedocs.io/en/stable/, accessed on 1 March 2021) was used to remove low-quality reads containing adaptor contamination, low-quality bases, and undetermined bases. FastQC (0.10.1) (https://www.bioinformatics.babraham.ac.uk/projects/fastqc/, accessed on 1 March 2021) was then used to verify sequence equality. Then, the murine genome was mapped by Bowite2 and Tophat2 [41,42]. Finally, all transcriptomes were merged and their expression levels were estimated by calculating the fragments per kilobase per million reads, using StringTie and Ballgown [43,44].

### 2.4. lncRNA Identification

First, the transcripts were discarded if they overlapped with known mRNAs and were shorter than 200 bp in length. Then, CPC0.9-r2 (https://cpc2.cbi.pku.edu.cn/, accessed on 1 March 2021) with default parameters (cpc2 -I novel.fa -o cpc2. out) and CNCI2.0 (https://github.com/www-bioinfo-org/CNCI, accessed on 1 March 2021) with default parameters (CNCI.py-f novel.fa-o CNCI.result-p 1-m ve-g novel.gtf-d genome.fa) were used to predict transcripts with coding potential [45,46]. Finally, transcripts of CPC scores < −1 and CNCI scores < 0 were removed. The rest of the transcripts were considered to be lncRNAs.

### 2.5. RNA Expression and Functional Analysis

The R package Ballgown with log2(fold change (FC >1)) or log2(FC) <−1 was used to screen DE lncRNAs and DE mRNAs with statistical significance *p* < 0.05. The function of DE mRNAs was analyzed using Gene Ontology (GO, https://geneontology.org, accessed on 1 March 2021) and the Kyoto Encyclopedia of Genes and Genomes (KEGG, https://www.kegg.jp/kegg, accessed on 1 March 2021); *p* < 0.05 represents statistical significance.

### 2.6. Target mRNA Prediction and Functional Analysis of lncRNAs

The function of lncRNAs was predicted by cis-targeted mRNAs of lncRNAs. In this study, coding genes 100 kbp upstream and downstream were selected via a Python script [47]. Co-expression analysis of DE lncRNAs and DE mRNAs was used to analyze the function of lncRNAs. The correlations between DE lncRNAs and mRNAs were examined via Pearson’s correlation analysis. Then, the functional analysis of the lncRNAs’ target mRNAs was demonstrated by BLAST2GO [48].

### 2.7. GO and KEGG Enrichment Analysis

GO terms and KEGG pathway enrichment analysis were performed to investigate the biological process, understanding the main functions of DE mRNAs and lncRNAs between E0.5 and E4.5, and then between E15.5 and E18.5.

### 2.8. Similarities and Differences in mRNAs and lncRNAs between the Two Stages

DE mRNAs, known DE lncRNAs, and novel DE lncRNAs were analyzed by Venn diagram to determine the similarities and differences between the embryo implantation stage and the labor stage, using OmicStudio tools (https://www.omicstudio.cn, accessed on 5 February 2022).

### 2.9. Murine Uterine Smooth Muscle Cell Isolation and Culture

Each of 10 female mice (6–10 weeks old) was injected with 0.1 mL eCG (equine chorionic gonadotropin, 200 U/mL) at 0 h and 24 h. After 48 h, the uterus was collected and placed into a 15 mL tube with 10 mL of PBS (0.2% penicillin–streptomycin, HyClone, Logan, UT, USA). Uteri were washed 3 times with PBS, had all residual adipose or connective tissue removed, and then were transferred to a new 10 cm culture dish and washed a further 3 times with PBS. Uteri were cut longitudinally to expose the uterine lumen and transferred into a new centrifuge tube with PBS. The cut uteri were scraped with a cell scraper to remove inner cells, such as murine endometrial epithelial cells and stromal cells. Then, the uteri were washed 3 times with fresh PBS in a new 50 mL tube with 10 mL of trypsin (2.5%, HyClone, Logan, UT, USA) added and incubated for 30 min at 37 °C while shaking (100 rpm). Ten milliliters of DMEM/F12 cell culture medium (10% FBS, HyClone, Logan, UT, USA) was added to stop incubation, and murine uterine smooth muscle (MSM) cells were collected by passing cell suspensions through a 100 µm nylon mesh for the first step, and then a 40 µm nylon mesh for the second step. The cell suspension was centrifuged at 1000 g for 8 min and then washed twice with PBS. The undigested uteri were collected and washed with PBS and then incubated again, as previously described.

### 2.10. P4 Treatment of MSM Cells

MSM cells were cultured in 6 pooled cell dishes separately. When the cell density was approximately 80%, the cells were treated with 100 nM P4 (Solarbio, Beijing, China) in DMSO for 24 h, while an equal volume of DMSO was used to treat the control group. Then, the cells were collected for qRT-PCR. Each experiment was replicated three times.

### 2.11. Validation of LncRNA Expression by qRT-PCR

For qRT-PCR analysis, we randomly selected two upregulated lncRNAs and two downregulated lncRNAs on E0.5 and E18.5 that represented differential expression levels from the RNA-Seq analysis of MSM cells treated with P4. Then, qPCR was performed on a two-step real-time PCR system (Abm, Vancouver, BC, Canada) with SYBR Green PCR Master Mix (Abm, Vancouver, BC, Canada). *GAPDH* was used as the control, and the 2^−∆∆Ct^ method was used to calculate the relative expression levels of lncRNAs and mRNAs. The expression of lncRNAs and mRNAs was examined with three independent biological replicates. The primers for *GAPDH* and lncRNAs are shown in Appendix A.

### 2.12. Statistical Analysis

All of the described experiments were carried out at least three times. *T*-test with SPSS 19.0 software (Version X, IBM, Armonk, NY, USA) was used to analyze the data between the DMSO group and the P4 group. Data are expressed as the mean ± SD of three independent processes. A *p*-value of < 0.05 was considered significant.

## 3. Results

### 3.1. Sequencing of RNA and Identification of lncRNAs and mRNAs in the Murine Uterus

To identify DE lncRNAs and mRNAs related to P4 levels in the murine uterus, cDNA was generated from three E0.5 samples (m0051, m0052, and m0053), three E4.5 samples (m0451, m0452, and m0453), three E15.5 samples (m151, m152, and m153), and three E18.5 samples (m181, m182, and m183). Means of the Illumina HiSeq 4000 Platform were used to obtain raw reads. Approximately 11.17 GB of data per uterus sample was retained after removing low-quality sequences and adaptor sequences. The GC content was approximately 48%. More than 91.12% of clean reads, including 75.98% uniquely mapped reads, were mapped to the murine genome using TopHat; these detailed data are shown in Appendix A.

In this study, the DE lncRNAs and mRNAs among the four groups were analyzed using edge R software with a set filter of |log2(FC)| ≥1 and *p* < 0.05. In our results, 1971 mRNAs, 493 known lncRNAs, and 1041 novel lncRNAs were found to show significant differential expression between E0.5 and E4.5, and 1149 DE mRNAs, 192 known DE lncRNAs, and 218 novel DE lncRNAs between E15.5 and E18.5 were found. At E0.5, 818 mRNAs, 232 known lncRNAs, and 567 novel lncRNAs were upregulated, while 1153 mRNAs, 261 known lncRNAs, and 103 novel lncRNAs were significantly downregulated (Figure 1D,E,F). At E18.5, 623 mRNAs, 89 known lncRNAs, and 83 novel lncRNAs were significantly upregulated, while 526 mRNAs, 103 known lncRNAs, and 135 novel lncRNAs were significantly downregulated, as indicated by boxplots of mRNA and lncRNA expression (Figure 1D,E,F). Additionally, DE mRNAs and lncRNAs are shown as a heatmap and volcano plot (Figure 2).

### 3.2. Enrichment Analysis of DE lncRNAs’ Target mRNAs

The main functions of DE lncRNAs’ target mRNAs were analyzed by GO. DE lncRNAs’ target mRNAs were enriched in GO terms, with functional annotation information. There were GO terms significantly enriched in the GO results that met the criteria of *p* < 0.05. The significantly enriched GO terms included membrane, integral component of membrane, and plasma membrane (Figure 3A,B,D,E). KEGG pathway analysis revealed 20 significantly enriched pathways (*p* < 0.05), including the PI3K-Akt signaling pathway, MAPK signaling pathway, and cytokine–cytokine receptor interaction (Figure 3C,F).

The DE lncRNAs’ target mRNAs were predicted by cis to the function of lncRNAs between the two periods. Similar to mRNAs, 197 GO terms with functional annotation information were enriched. Based on the biological process analysis, the DE lncRNAs’ target mRNAs were involved in membrane, nucleus, and G-protein-coupled receptor activity (Figure 3A,B,D,E). The results of the KEGG pathway enrichment analysis suggested that these DE lncRNAs’ target mRNAs were mainly significantly connected with viral carcinogenesis and phagosome and cytokine–cytokine receptor interactions (*p* < 0.05) (Figure 3C,F).

### 3.3. Similar DE mRNAs and DE lncRNAs between the Two Stages

To verify which mRNAs and lncRNAs were similar between the two stages, Venn diagrams were used to visualize the similarities. A total of 117 DE mRNAs, 19 known DE lncRNAs, and 31 novel DE lncRNAs were found to be similar (Figure 4A,B,C). It is worth noting that most of the DE lncRNAs were intron lncRNAs.

### 3.4. qRT-PCR Analysis of mRNA Expression in MSM Cells after P4 Treatment

To evaluate whether lncRNA expression was influenced by P4, the expression of four lncRNAs was validated by qRT-PCR (Figure 4D). Compared with those in the DMSO subgroup, the expression levels of lncRNA-MMP11 were significantly downregulated in the P4 treatment subgroup, while lncRNA-ANKRD37 was significantly upregulated (Figure 4D); however, the expression of lncRNA-LAMA1 and lncRNA-RPL24 was not significantly affected (Figure 4D).

## 4. Discussion

The uterus is one of the most important organs for female mammalian reproduction. The function of the uterus is regulated by P4- and E2-mediated cell proliferation and differentiation, and P4 is considered to be the main hormone involved throughout the whole pregnancy [9,13,14,49]. Interestingly, serum P4 levels fluctuate significantly during pregnancy. P4 becomes completely dominant at E4 to be receptive, and becomes refractory to implantation at E5 [50,51], while the dominance of P4 begins to change at E15.5, and reaches a low point at E18.5 with the increased expression levels of some miRNAs, such as miR200 families [27,52]. Embryo implantation and labor are highly complicated and multifactorial processes. Making better sense of the successful molecular dialog between mother and embryo during implantation and labor may improve the understanding of the causes of pregnancy failure [53,54]. Consequently, finding potential mRNAs and lncRNAs regulated by P4 is important in order to characterize molecular mechanisms and thereby provide a basis for understanding implantation and labor. This can also enable the development of molecule-based drugs that prevent or treat implantation failure, pregnancy loss, and preterm labor. Our work revealed the expression of DE mRNAs, known DE lncRNAs, and novel DE lncRNAs in the murine uterus during implantation and labor. A large number of lncRNAs were found in the murine reproductive tracts, oocytes, and two-cell embryos to morula, while the lncRNAs were considered to play important functional roles in oocyte and embryonic development, pre-implantation, spermatogenesis, and some other reproductive processes [55,56,57]. In the present study, 493 known significantly DE lncRNAs were identified in murine uteri between E0.5 and E4.5, and 192 known lncRNAs were identified in murine uteri between E15.5 and E18.5 (Figure 1E), while 1041 novel DE lncRNAs were found between E0.5 and E4.5, and 218 novel DE lncRNAs were found between E15.5 and E18.5 (Figure 1F). A total of 117 common DE mRNAs, 19 common known DE lncRNAs, and 31 common novel DE lncRNAs were identified from the two stages. Most of the lncRNAs were intron lncRNAs, suggesting that the lncRNAs mainly take place in intergenic regions to influence active transcription in the murine uterus during embryo implantation and labor (Figure 4B). Similar to previous studies, the expression levels and conservation of putative lncRNAs identified in the present study were lower than those of protein-coding genes. Simultaneously, the lncRNAs have shorter transcript lengths, smaller exon numbers, and shorter ORF lengths than protein-coding genes. Notably, lncRNA-ANKRD37 has a much higher expression level than other lncRNAs, and even some mRNAs.

In the present study, the cis and trans lncRNA targets were found to be mainly enriched in cellular processes from the two stages by GO terms. These lncRNAs may play important roles in embryo implantation and labor. Furthermore, functional analysis of the cis lncRNA targets was also significantly enriched in the defense response. These results indicate that lncRNAs participate in the immune response by regulating neighboring protein-coding genes. The GO terms of upregulated lncRNA targets in cis were related to cellular component biogenesis and immune function at E0.5, and E4.5 would be helpful for embryo implantation. Similarly, the target genes of upregulated lncRNAs were related to immune function at E18.5, resulting in labor. Furthermore, these DE lncRNAs’ target mRNAs at E15.5 and E18.5 were found to be enriched in the D-glutamine and D-glutamate metabolism pathways by KEGG analysis, which has been reported during late pregnancy [58]. As shown in Figure 1D–F, the numbers of DE mRNAs and DE lncRNAs are higher during the embryo implantation stage than the labor stage, which might be due to the need for more complex regulation in embryo implantation. GO scatterplot results show that the lncRNAs’ target mRNAs are related to dehydrogenase activity, citrullination, and phosphorylation at E0.5 and E4.5, while at E15.5 and E18.5 they are related to nucleosome, iron correlation reaction, and regulation of kinase activity (Figure 3B,E). It might be the case that the labor stage requires more frequent iron interactions to contract, and expends much more energy than implantation.

Although some of these lncRNAs have been reported in other cells, the relationship between DE lncRNAs and the two periods has not been demonstrated. MiR-491-5p and miR-214-3p were sequestered by the lncRNA VPS9D1AS1, which elevated GPX1, promoting cell proliferation in acute lymphoblastic leukemia [59]. The expression of the lncRNA SNHG1 was high in cervical cancer cell lines and cervical cancer tissues, while knockdown of the lncRNA SNHG1 can relieve cell migration, proliferation, and invasiveness in HeLa and C33A cells [60]. LncRNA SNHG1 can also regulate BCL-XL by modulating miR-140-5p to reduce cell apoptosis and recover cell viability in SH-SY5Y cells [61]. LncRNA MIR99AHG activated the FOXP1-mediated Wnt/β-catenin pathway by competing for endogenous RNA of miR-577, promoting gastric cancer [62]. It is interesting to note that the lncRNA MIR99AHG could promote autophagy by binding to ANXA2, then generate miR-99a to suppress the expression of mTOR, postponing lung adenocarcinoma progression [63]. The lncRNA DNM3OS sponged miR-126 to affect cell proliferation and apoptosis by regulating IGF1, by binding to the promoter of miR-126 in CHON-001 cells [64]. The expression level of the lncRNA HAND2OS1 was higher on day 4 of pregnancy than day 1 of pregnancy, while the level increased dramatically on day 6, which is similar to the findings of this study, where two transcripts of the lncRNA HAND2OS1 were downregulated at E0.5 [65]. Ankyrin repeat domain protein (ANKRD) is a large family of proteins including enzymes, transcription factors, and membrane receptors, which are composed of multiple copies of ankyrin repeat (AR) with a 33-amino-acid motif [66]. ANKRD37 is known to be degraded by Fem1b through the ubiquitin system [67]. The translocation of ANKRD37 into cell nuclei is essential for autophagy, inducing p62 expression and LC3-I/LC3-II conversion, while autophagy has also been shown to be associated with embryo implantation and labor [68,69,70]. There has been no study of ANKRD37 in the murine uterus. In the present study, lncRNA-ANKRD37 was upregulated by P4 treatment, indicating that its function of uterine quiescence may be related to autophagy during pregnancy.

This study characterized the similar and differential expression of lncRNAs and mRNAs between the embryo implantation stage and the labor stage. Most importantly, 19 common known DE lncRNAs and 31 common novel DE lncRNAs were found between the two stages, and these lncRNAs might be regulated by P4. In the present study, lncRNA-ANKRD37 was considered a vital molecule to study further, due to its high expression level and high expression during the implantation stage, labor stage, and P4 treatment of MSM cells. These findings suggest that lncRNAs may play a role in murine uterine function. This synergism between lncRNAs and mRNAs is critical for uterine function under direct or indirect P4 regulation. Further studies of this crosstalk may reveal novel functions of lncRNAs in murine embryo implantation and labor, and the findings of such studies will be helpful for the research and development of drugs and the precise regulation of mammalian reproduction.

## 5. Conclusions

In conclusion, 117 common DE mRNAs, 19 common known DE lncRNAs, and 31 common novel DE lncRNAs were found between the embryo implantation stage and the labor stage. These DE lncRNAs may have similar functions in the two stages. The lncRNA-ANKRD37 was considered a vital molecule to study further due to its high expression level.

## Figures and Tables

**Figure 1 animals-12-00399-f001:**
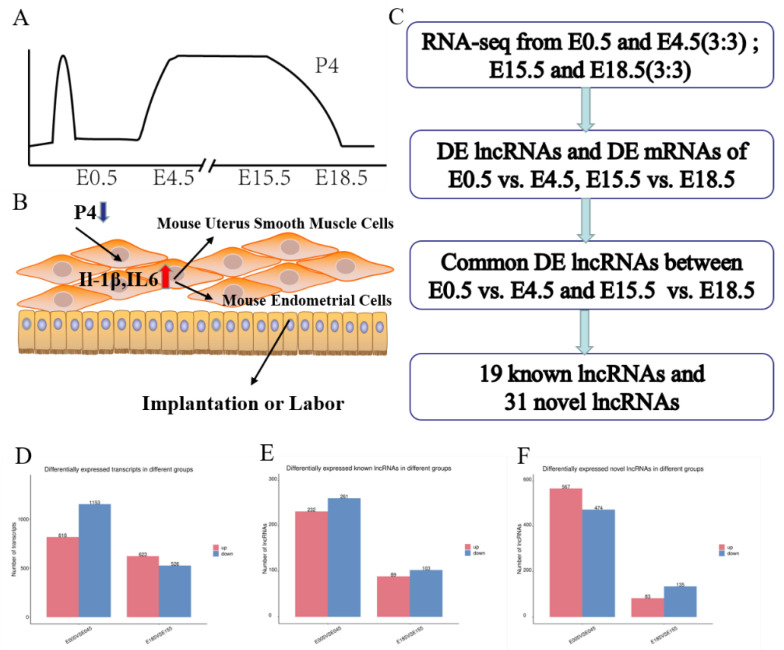
Identification of P4-associated lncRNAs: (**A**) Windows of uterine P4 regulation throughout the whole pregnancy. (**B**) Effects of P4 on implantation or labor. (**C**) The screening process used in this study. (**D**) The number of DE mRNAs between the two stages; the red boxes denote upregulated mRNA numbers, while the blue boxes denote downregulated mRNA numbers. (**E**) The numbers of known DE lncRNAs between the two stages; the red boxes denote upregulated known lncRNA numbers, while the blue boxes denote downregulated known lncRNA numbers. (**F**) The numbers of novel DE lncRNAs between the two stages; the red boxes denote upregulated novel lncRNA numbers, while the blue boxes denote downregulated novel lncRNA numbers.

**Figure 2 animals-12-00399-f002:**
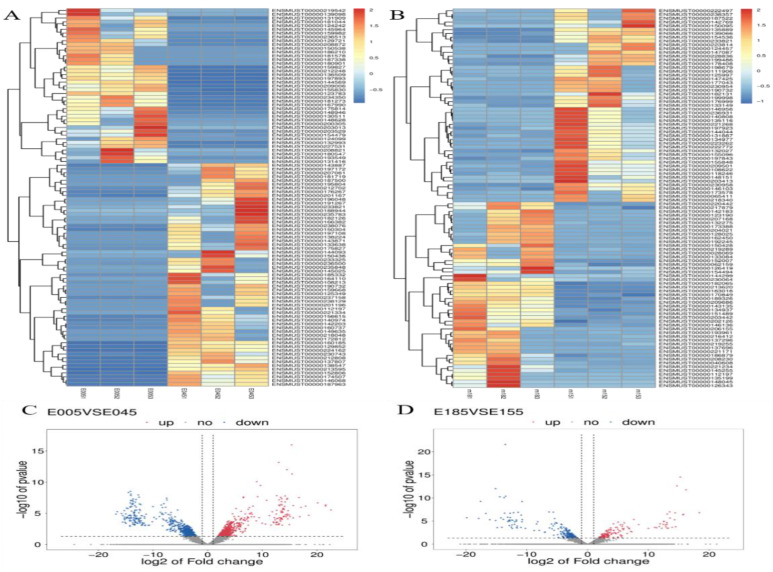
The differential expression of lncRNAs between E0.5 and E4.5, and between E15.5 and E18.5: (**A**) Heatmap of lncRNAs between E0.5 and E4.5. (**B**) Heatmap of lncRNAs between E15.5 and E18.5. (**C**) Differential expression of lncRNAs between E0.5 and E4.5. (**D**) Differential expression of lncRNAs between E15.5 and E18.5.

**Figure 3 animals-12-00399-f003:**
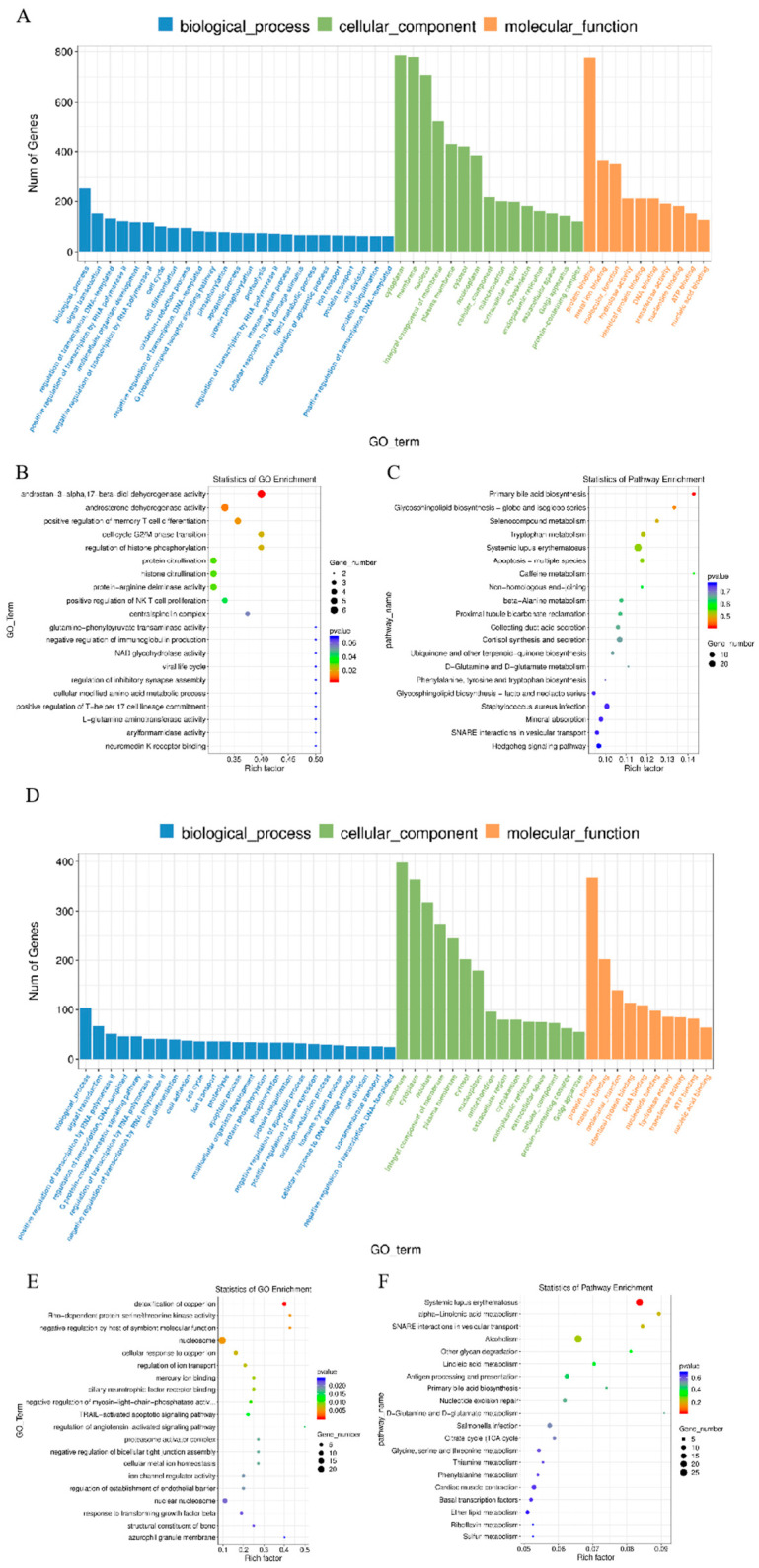
GO and KEGG analysis of DE lncRNAs’ target mRNAs’ expression: (**A**) Histogram of GO enrichment of DE lncRNAs’ target mRNAs between E0.5 and E4.5. (**B**) Scatterplot of GO enrichment for DE lncRNAs’ target mRNAs between E0.5 and E4.5. (**C**) Scatterplot of KEGG enrichment for DE lncRNAs’ target mRNAs between E0.5 and E4.5. (**D**) Histogram of GO enrichment of DE lncRNAs’ target mRNAs between E15.5 and E18.5. (**E**) Scatterplot of GO enrichment for DE lncRNAs’ target mRNAs between E15.5 and E18.5. (**F**) Scatterplot of KEGG enrichment for DE lncRNAs’ target mRNAs between E15.5 and E18.5.

**Figure 4 animals-12-00399-f004:**
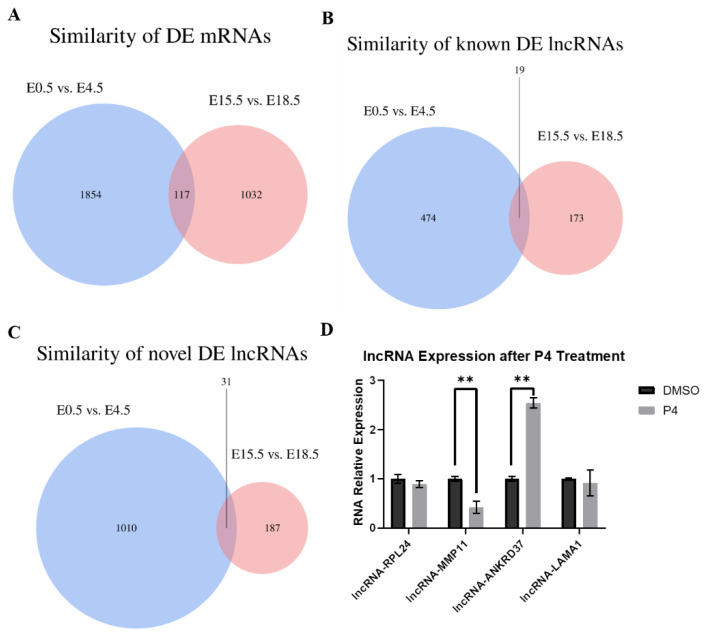
Venn diagram showing DE mRNA and DE lncRNA expression, and qRT-PCR results: (**A**) Similarities in DE mRNAs between E0.5 and E4.5, and between E15.5 and E18.5. (**B**) Similarities in known DE lncRNAs between E0.5 and E4.5, and between E15.5 and E18.5. (**C**) Similarities in novel DE lncRNAs between E0.5 and E4.5, and between E15.5 and E18.5. (**D**) qRT-PCR results for P4 treatment on MSM cells. ** *p* < 0.01. Data are presented as the mean ± SEM.

## Data Availability

The clean library sequencing data reported in the research have been uploaded to Gene Expression Omnibus (GEO) with the accession no. GSE195795 (https://www.ncbi.nlm.nih.gov/geo/query/acc.cgi?acc=GSE195795, accessed on 1 February 2022).

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
