# Peer review of "Comparison of lncRNA Expression in the Uterus between Periods of Embryo Implantation and Labor in Mice"

_animals, 2022, doi:10.3390/ani12030399_

Round 1

Reviewer 1 Report

In this paper, Zhao and colleagues analyzed the expression of mRNA and lncRNA during mouse embryo implantation and labor.  Using pregnant mice at E0.5, E4.5, E15.5, and E18.5, they found 1971 differentially expressed (DE) mRNAs and 1534 DE lncRNAs between E0.5 and E4.5 stages., and 1149 DE mRNAs and 410 DE lncRNAs between E15.5 and E18.5 stages. Moreover, they found that 117 DE mRNAs and 19 DE lncRNAs were commonly expressed between the two stages, indicating that their expression may be regulated by P4.

- In the abstract, the authors should add their results obtained with MSM cells; - Did the authors eliminate the embryos from uterus? - Why did the authors not performed a DNase I treatment, to eliminate DNA contamination? - Did the authors characterize the isolated MSM cells? - Why did the authors not compare the obtained results between E0.5/E4.5 and E15.5/E18.5?

Minor concerns:

- In line 14, the meaning of P4 and E2 should be specified; - In line 129, please substitute “killed” with another word; - In line 155, GAPDH should be italicized.

Reviewer 2 Report

A manuscript entitled “Comparison of lncRNAs expression in uterus between periods of embryo implantation and labor in mouse” by Zhao et al, submitted to Animals, describes differentially expressed lncRNA and mRNA on days 0.5, 4.5, 15.5 and 18.5 in mice. The main emphasis of this manuscript is differential expression of uterine lncRNAs at important pregnant days as they are influenced by progesterone (P4) concentrations. It would thus be much better if P4 concentrations were measured at corresponding days, which directly tie those to lncRNA expression.

Most important point in this manuscript is whether lncRNAs found were unique/novel or those already identified and described elsewhere. In this regard, description of differentially expressed (DE) mRNAs should be minimized, unless those found in the present study are novel, not already found and described elsewhere.

The manuscript should be prepared in a much more careful manner. For example, but not limited to, line 86, the sentence starts with RNA. This is ribosomal RNA thus the sentence should be like “Total RNA was depleted of ribosomal RNA (rRNA) molecules using the Robo-Zero rRNA Removal kit, according to the manufacturers’ instructions”. In this manuscript, identification of lncRNA from mRNAs is crucial, therefore CPC and CNCl (line 102) should be described very carefully.

Line 130, A sentence, “…and the uterus was placed in a clean bench into a 15-mL tube…”, requires an attention: “in a bench” should be placed before “the uterus”, or “in a bench” can be removed.

Line 132, this reviewer believes that the authors want to say as follows: Uteruses were washed 3 times with PBS, all residual adipose or connective tissue removed, and then transferred to a new 10-cm culture dish…

Line 136, the sentence should read: …such as mouse endometrial epithelial and stromal cells.

Line 154, Two (one space) step Real-Time PCR system.

Line 161, remove “and” before t-test.

Line 163, Remove “and p<0.01 was considered to be a very significant difference.”

Fig. 3 needs to be re-organized and carefully presented. Gene family description of Fig. 3A&D is inverted, which definitively corrected. Usually, the presentation should be from left to right and top to bottom. If this is not possible, the data should be presented in a way the readers can follow/examine easily.

Fig. 4 requires attention, too. A title of Fig. 4 should be “Venn diagram showing DE mRNA and DE lncRNA expression and qRT-PCR results”. A label on top of Venn diagram should be E005 vs. E045, not E005VSE045. The remaining three labels require the same attention. This reviewer does not understand why the authors opted to present the data as is. Venn diagram should be the comparison between those of E005 and E045, and the others with those of E155 and E185, rather than the present ones. However, if the authors intention was to compare those of E005 and E045, P4 increment, and those of E155 and E185, P4 decline, then the comparison should reflect this and the presentation should be done accordingly.

Discussion:

Lines 234-236, changes in P4 concentrations should be presented in a way that shows exactly what the authors want to show. For examples, the sentence, “Specifically, the serum P4 level briefly increases and then rapidly decreases during the implantation stage”, is not correct.

Lines 243-244, “Our work revealed the differentiation…”, the data and their explanation do not match, thus this needs to be corrected accordingly.

The remaining portions of Discussion are based on mRNA data, which have been studied and described elsewhere. Thus, Discussion should be re-written so that the contents represent what the authors studied.

Round 2

Reviewer 1 Report

The Authors responded to the raised issues, improving the quality of their MS. Is it suitable for acceptance in Animals in this form. 

Reviewer 2 Report

The revised manuscript entitled “Comparison of lncRNAs expression in uterus between periods of embryo implantation and labor in mouse” was again submitted to Animals. This time, the authors’ intension (research objective), the experimental results and the way manuscript written are all in agreement, demonstrating lncRNA play important roles during two critical periods in pregnancy, implantation and labor. In particular, their lncRNA are clearly divided into those of known and the others novel molecules. This reviewer believes that this was the authors’ objective, supported by the valuable data.

Thus, this reviewer highly recommends this manuscript for publication in Animals.